# Time from treatment initiation to HIV viral suppression in public care facilities in Brazil: A nationwide linked databases cohort

**Maria Ines Battistella Nemes**[1]*, **Ana Paula Sayuri Sato**[2], **Barbara Reis-Santos**[1], **Ana Maroso Alves**[1], **Felipe Parra do Nascimento**[1], **Bruce Agins**[3]

**1** Departament of Preventive Medicine, School of Medicine, University of São Paulo, São Paulo, São Paulo, Brazil, **2** Department of Epidemiology, School of Public Health Universidade de São Paulo, University of São Paulo, São Paulo, São Paulo, Brazil, **3** Division of Epidemiology, Institute for Global Health Sciences, University of California, San Francisco, San Francisco, California, United States of America

* mibnemes@usp.br

## Abstract

### Objectives

To analyze the time between antiretroviral therapy (ART) initiation and the first HIV viral load (VL) test <40 copies—time to suppression (TS)—in a cohort of persons aged ≥15 years, between 2015–2018 in outpatient HIV care facilities of the Brazilian Unified Health System, as well as to analyze whether individual and facility characteristics accelerate or delay TS.

### Methods

This was a cohort study with data from a linkage of national HIV databases, following a previously published procedure. Two types of variables were examined: individual-level (sex, age group, race/skin color, education, baseline CD4 cell count and VL, initial ART regimen, adherence, ART regimen change and number of VL tests until suppression) and facility-level (national and metropolitan region, caseload). Multilevel parametric accelerated failure time survival models were used. Fixed and random effects were analyzed through null, sociodemographic, combined sociodemographic and clinical, and facility-related variables, adjusted for the number of VL tests until suppression. Likelihood, interquartile range, and proportion of change in variance were used for comparisons.

### Results

Of 132,540 participants, 89.4% (114,696) achieved viral suppression: 20.8% within three months, and 56.4% within six months. Median TS was 161 days, varying from 31 to 1,426 days, depending on the time interval between initiation and VL testing. Among those who had VL testing within 66 days, median TS was 55 days. All individual and facility-related variables were associated with TS, explaining the 16.2% and 13.2% variability, respectively.

**Data Availability Statement:** All data underlying the findings described in the manuscript are fully available without restriction. All files are available

from the TimeToSupressin_dataset database on https://figshare.com/ via DOI: 10.6084/m9.figshare.26314996

**Funding:** CNPq (Conselho Nacional de Desenvolvimento Científico e Tecnológico | National Council for Scientific and Technological Development) - U$D 9,600.00 (MIBN) https://www.gov.br/cnpq/pt-br OPAS (Organização Pan-Americana da Saúde - Organização Mundial da Saúde - Região das Américas| Pan American Health Organization - World Health Organization - Americas Region)- U$D 31,670.00 (MIBN) https://www.paho.org/en Pró-Reitoria de Pesquisa da Universidade de São Paulo - USP | USP Research and Innovation Deanship University of São Paulo) – U$D 6,360.00 (APSS) https://sites.usp.br/prp/ The funders had no role in study design, data collection and analysis, decision to publish, or preparation of the manuscript.

**Competing interests:** The authors have declared that no competing interests exist

## Conclusions

This was the first Brazilian nationwide cohort to analyze TS. It is also one of the largest operational cohorts globally to assess healthcare facility characteristics. The findings indicated that both individual and facility-related characteristics contribute to TS. Strengthening VL monitoring should be included as part of a coordinated effort to improve the quality of care provided for people living with HIV/AIDS in Brazil.

## Introduction

The clinical and public health benefits of HIV viral load suppression (VS) through combination antiretroviral therapy (ART) are well-established, supported by robust scientific evidence [1–4] which forms the basis for policies and strategies that aim to engage the international community, governments, and civil society in achieving and maintaining VS targets. Cascade-based official reports from HIV national programs [5, 6] compile indicators that are immediately useful for healthcare professionals, HIV care facility managers, and system managers [7, 8]. These indicators are, however, limited by their cross-sectional design, as they assess the VS proportion of the group of persons in care within a fixed time interval (usually six, 12, or 24 months) between ART initiation and VS achievement, without considering temporal evolution [9]. Acknowledging these limitations, researchers have advocated that the parameter used by HIV programs to monitor ART success be time to VS [10–14].

With widespread use of integrase inhibitors [9], an increase has been seen in studies examining time to VS continuously [15–20]. As associated factors, a significant portion of these studies analyze individual characteristics, whereas fewer of them also analyze care facility characteristics [20–23].

Up to the time of writing this paper, we identified only one population-based cross-sectional study in Brazil that assessed the median time between ART initiation and VS, conducted from 2004 to 2018 in one Brazilian state [24]. No study including care facility characteristics in its analysis was, however, found.

This article aimed to analyze the time between ART initiation and first VS in a cohort of persons living with HIV/AIDS (PLWHA) who initiated ART between 2015 and 2018 in outpatient HIV care facilities of the Brazilian Unified Health System (SUS). A further objective was to analyze whether individual characteristics and healthcare facility characteristics accelerate or delay time to VS.

## Materials and methods

### Study design

This was an observational longitudinal study including PLWHA aged ≥15 years who are participants in the Qualiaids-Brazil Cohort.

### Context

In Brazil, the establishment of the national program for STD/AIDS in 1986 [25] prompted the implementation of hospital and outpatient facilities for the treatment of people with AIDS, mostly within pre-existing structures of the public health system [26], which officially became the SUS [27] in 1988. The spread of AIDS in the country and the introduction of ART in 1996, which turned HIV disease into a chronic condition, led to a significant expansion in the

number of outpatient facilities, reaching approximately 1,300 today [28]. These facilities are the exclusive providers of ART medication for approximately 852,000 people. Clinical monitoring of PLWHA can be carried out within the public or private healthcare system. According to the cohort database, it has been estimated that 69% of PLWHA received outpatient monitoring through SUS facilities [28, 29].

## Study population

The Qualiaids-Brazil Cohort comprises PLWHA who received treatment at SUS facilities that participated as respondents in a national survey regarding facility characteristics (Qualiaids Survey) [30–34]. PLWHA aged ≥15 years were included if they had their first ART dispensation between 2015 and 2018, a CD4 test conducted between 120 days before and 30 days after the first ART dispensation, and at least two viral load (VL) test results. The exclusion criteria considered inconsistencies in PLWHA data among the data sources used. Further information on the construction of the database and the profile of the Qualiaids-Brazil Cohort can be obtained from a previous publication [29].

For this study, only PLWHA who were in treatment for more than 30 days were included [35]. This was due both to the inability to assess treatment effect for very early suppressions and the possibility of delays in recording the first dispensation date. Those who had their last VL test in the SUS before their first ART dispensation were excluded.

## Follow-up

The date of the first medication dispensation was defined as the initiation of ART, and the date of the last VL test was set as the final follow-up date. Deaths and loss to follow-up were classified as censoring events. The mortality data comprised only those deaths attributed to HIV-related causes [29]. Loss to follow-up was attributed to individuals who went more than 100 days without picking up medication or having VL tests until the final follow-up date of the cohort (December 31, 2018) [29]. Administrative censoring referred to individuals who did not achieve VS by the end of the follow-up period.

## Study variables

The dependent variable was the time from treatment initiation to VS, defined as the number of days elapsed between the date of the first ART dispensation and the date of the first VL <40 copies/mL after ART initiation. The independent variables included data on individuals and treatment facilities.

The following sociodemographic characteristics of individuals were included: *sex* (female; male), *age group* (15–19; 20–29; 30–39; 40–49; 50–59; ≥60 years), *race/skin color* (white; black; yellow; mixed-race; indigenous), and *education level* (none; 1–3; 4–7; 8–11; ≥12 years). For the variables *race/skin color* and *education level* 7.3% and 18.6% of entries were missing data respectively. Therefore, imputation was performed to reduce potential biases in the analyses by using Stata 15's *mi impute mlogit* library, which employs a multinomial logistic regression model for imputation of nominal variables. The variable *transmission category* was excluded due to the high proportion of missing records (34.2%) in the original database.

Among individual clinical characteristics, *baseline CD4 lymphocyte count* showed considerable variation within the interval between the date of the test and treatment initiation, as has been reported elsewhere in the literature [36, 37]. We thus decided to consider as baseline the CD4 lymphocyte count performed between six months before and 30 days after the treatment initiation date (<200; 200–349; 350–499; ≥500 cells/mm$^3$). The remaining clinical variables included were: *initial VL count* (≤100,000; >100,000 copies/mL) *and occurrence of active*

*tuberculosis episode until the first VS* (no; yes). Regarding treatment, the variable *initial therapeutic regimen* comprised ART medications specified in the clinical protocol of the period: preferred regimens–non-nucleoside reverse transcriptase inhibitor + 1 integrase inhibitor (NNRTI+1INI) and nucleoside reverse transcriptase inhibitors + 1 non-nucleoside reverse transcriptase inhibitor (2NRTI+1NNRTI); special regimens authorized by regional technical chambers and unauthorized regimens [38].

Additionally, the following variables were included: *change in therapeutic regimen until VS* (no; yes), *adherence to treatment until the first VS*, measured by the Medication Possession Ratio - MPR (<80%; 80–94%; ≥95%), and the *number of VL tests conducted until VS*.

The variables regarding treatment facility characteristics were: *facility location according to geographical region* (North; Northeast; Central-West; South; Southeast), *metropolitan region* (no; yes), and *total number of PLWHA who initiated ART at the facility between 2015–2018* (≤50; 51–500; >500).

## Data analysis

The dependent variable was the distribution of the time from treatment initiation to VS, for which Normal, Log-Normal, Weibull, Exponential, and Gamma distributions were tested [39]. The Log-Normal distribution showed the best fit to the data. We therefore decided to use a parametric multilevel accelerated failure time survival model with a Log-Normal distribution for the analysis of cohort data [40, 41].

Initially, the absolute and relative frequencies of the qualitative variables were examined, and measures of central tendency and dispersion were calculated for the dependent (quantitative) variable. Subsequently, bivariate analyses were conducted between the dependent variable and each individual and facility-related variable. Finally, these variables were analyzed in multiple hierarchical models. The two levels of aggregation in the models (individual and facility-related) and the hierarchy of variables were defined based on the theoretical model presented in Fig 1.

Four models were constructed for the multiple analysis, aiming to measure the fixed and random effects of the variables at each level. Based on the null model, individual variables were included in the following sequence: sociodemographic characteristics, disease severity, and

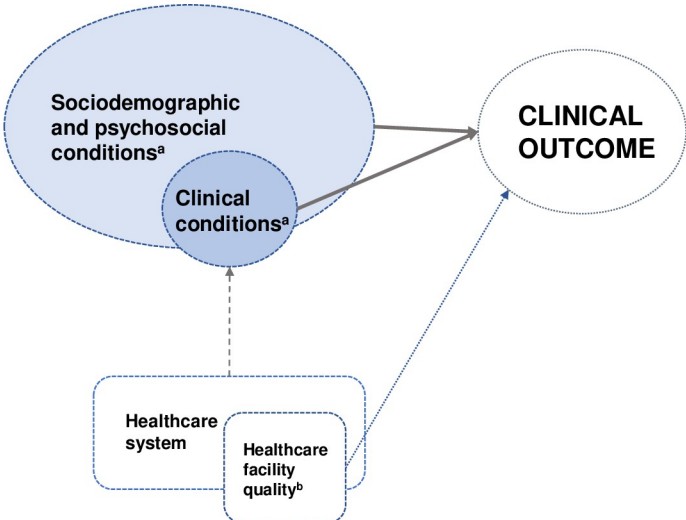

**Fig 1. Theoretical model of clinical outcome production in HIV infection treatment.**

treatment. The model relative to treatment facilities included only the block of facility-related variables. Comparisons between the models were conducted using likelihood, interquartile range, and the proportion of change in variance. The results were presented as time ratios (TR) and their respective 95% confidence intervals (95% CI). The significance level used for all analyses was 5%.

For the characterization of random effects, the four analyzed models were compared in terms of random variance (standard error-SE), intraclass correlation coefficient (ICC), log likelihood, and proportional change in variance (PCV) [42–45].

The PCV, at the contextual level, measures the variation of time to VS across healthcare facilities and is defined as [42, 43, 45]:

$$PCV = \frac{\text{Random variance of the null model} - \text{random variance of the final model}}{\text{Random variance of the null model}}$$

Contextual differences in time to VS can be attributed to random effects or differences in individual composition of the facilities in terms of sociodemographic factors, disease severity, or treatment initiation of the patients. Therefore, the PCV was calculated by comparing the null model with the other models that included only individual variables (models 1 and 2) and only facility-related variables (model 3).

The ICC is the proportion of the total variance of time to VS among treatment facilities that is attributed to facility variability; in other words, it measures the heterogeneity of time to VS attributed to differences between facilities, and is calculated for each model using the formula [44]:

$$ICC = \frac{\text{Random variance}}{\text{Random variance} + \text{individual variance}}$$

As detected in observational [24, 46–50] studies, there has been variation in adherence to the VL monitoring protocol, which may affect the estimates [46–51]. To minimize this limitation, all models were adjusted for the number of VL tests received by the individual.

Additionally, a sensitivity analysis was conducted by replicating the models with the inclusion of the variable *transmission category*, using the subset of individuals with complete information for this variable (S2 Table), and without adjustment for the number of VL tests received (S3 Table). All analyses were performed using Stata 15.0's *mestreg* package.

## Ethical considerations

The project was approved by the Ethics Committee on Research on Human Beings of the USP School of Medicine on January 23, 2020 (CAAE: 27659220.3.0000.0065; Opinion: 3.807.435), following the recommendations of Resolution No. 466 of 2012 from the Brazilian National Health Council. The data was sent to the research team on May 10, 2020. There were no variables or information that could identify the individuals in the data set received.

## Results

The Qualiaids-Brazil Cohort comprises 132,540 PLWHA who initiated ART in Brazil between January 1, 2015, and December 31, 2018. For this study, considering eligibility criteria, 128,360 PLWHA were included who had been in treatment for more than 30 days and received at least one VL test after starting ART. A total of 2,544 (1.9%) individuals who had been in treatment for fewer than 30 days, and 1,636 (1.2%) individuals who did not receive a VL test before the first ART dispensation were excluded. The censoring events were as follows: 1,496 (1.1%) deaths, 3,838 (2.9%) losses to follow-up, and 10,988 (8.3%) administrative censorings [29].

The distribution of time to VS is shown in Fig 2. Of the PLWHA considered in this study, 89.4% (114,696) achieved VS, with 20.8% (23,801) reaching it within 3 months after treatment initiation, 56.4% (64,636) within 6 months, and 87.5% (100,408) within 12 months.

The median time to VS for all included individuals was 161 days, ranging from 31 to 1,426 days, depending on the time interval between treatment initiation and the control VL test. Shorter intervals resulted in smaller medians (S1 Table).

Regarding the distribution of individual characteristics of the participants, higher proportions of PLWHA were identified as male (69.8%), aged 20–39 years (64.3%), with >8 years of schooling (62.8%), and of race/skin color brown or black (54.6%). A total of 26% of participants had an initial CD4 count <200 cells/mm$^3$, whereas initial VL >100,000 copies/mL was observed in 8.2% of participants. The initial therapeutic regimen was 2NRTI + 1NNRTI for 61.8% of participants; 3.2% changed their therapeutic regimen before the first VS. 42.5% of participants experienced an episode of tuberculosis before the first VS. 42.5% of participants had treatment adherence rates of ≥95%.

Regarding the treatment facilities, Fig 3 shows the distribution of the 941 included facilities and of PLWHA monitored in them, according to geographic region and location in metropolitan areas. The Southeast region accounted for most facilities (59.0%) and PLWHA (41.2%). There was a predominance of facilities located in capital cities or metropolitan regions (66.2%) and facilities assisting 51–500 individuals who started ART during the period (49.0%).

Table 1 shows the observed proportions of individual characteristics and the bivariate analyses between the dependent variable and each of the individual variables (sociodemographic, clinical, and treatment-related).

Table 2 shows the bivariate analysis between the dependent variable and each of the facility-related variables.

The bivariate analysis showed that PLWHA who received care in facilities in the North (TR = 1.13; 95% CI: 1.06; 1.21), Northeast (TR = 1.14; 95% CI: 1.09; 1.19), and Central-West

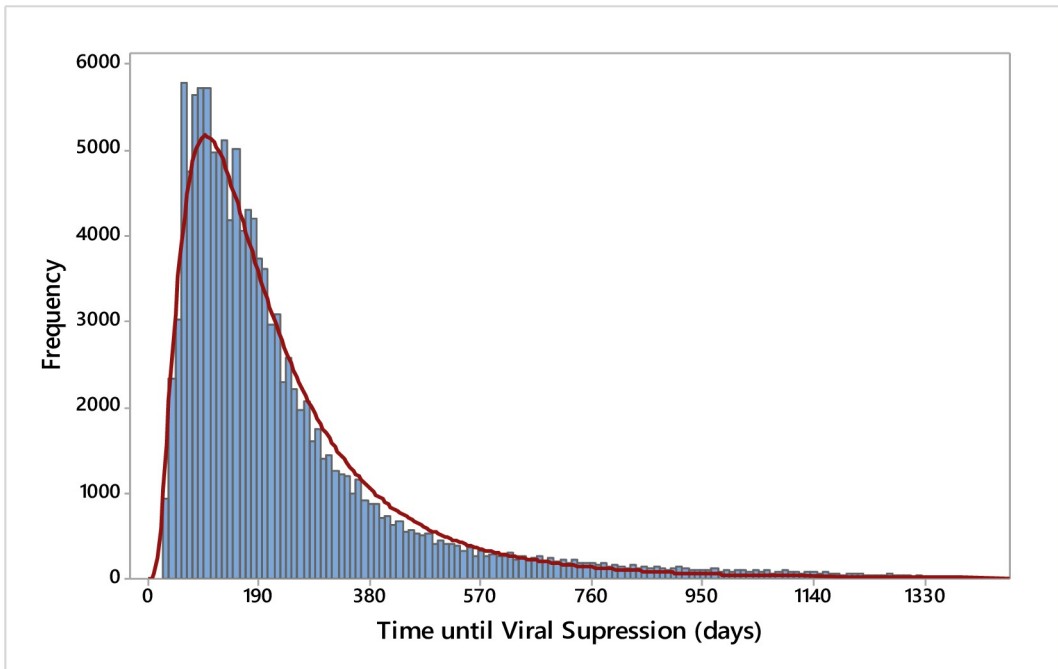

**Fig 2. Distribution of time to VS, in days, Qualiaids-Brazil Cohort, 2015–2018 (N = 128,360).**

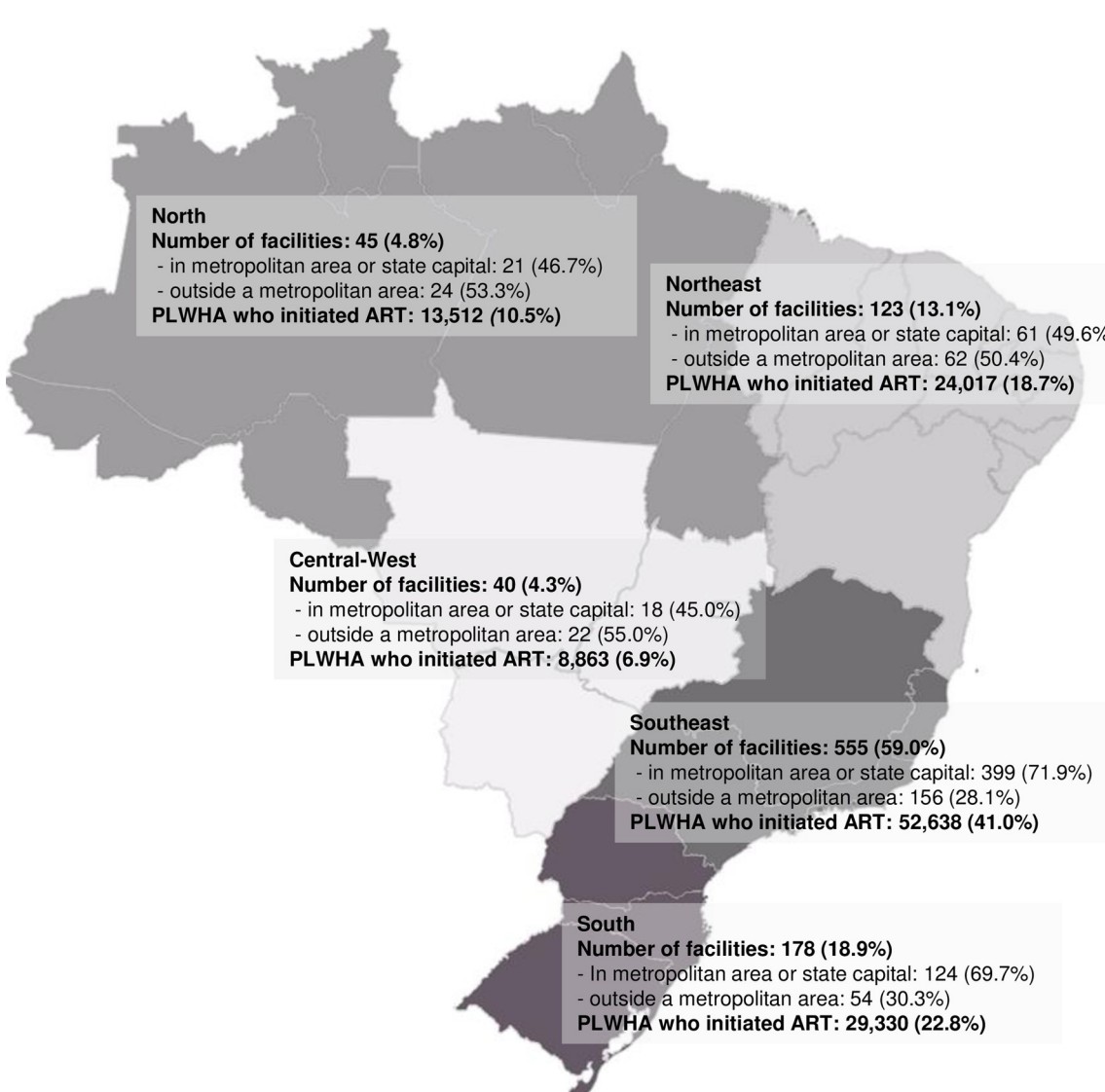

| Number of facilities ranked by size | Central-West | | Northeast | | North | | Southeast | | South | | Total | |
|---|---|---|---|---|---|---|---|---|---|---|---|---|
| | n | % | n | % | n | % | n | % | n | % | n | % |
| Up to 50 persons | 6 | 15.0 | 33 | 26.8 | 8 | 17.8 | 306 | 55.1 | 44 | 24.7 | 397 | 42.2 |
| 51 to 500 persons | 27 | 67.5 | 72 | 58.5 | 28 | 62.2 | 219 | 39.5 | 115 | 64.6 | 461 | 49.0 |
| More than 500 persons | 7 | 17.5 | 18 | 14.6 | 9 | 20.0 | 30 | 5.4 | 19 | 10.7 | 83 | 8.8 |
| Total | 40 | 100.0 | 123 | 100.0 | 45 | 100.0 | 555 | 100.0 | 178 | 100.0 | 941 | 100.0 |

**Fig 3. SUS HIV treatment facilities and number of individuals included in the cohort who initiated ART in 2015–2018, categorized by macro-region of the country.** Qualiaids-Brazil Cohort, 2022 (N = 941).

(TR = 1.10; 95% CI: 1.03; 1.18) regions had a longer time to VS compared to those in the Southeast. This finding was also observed for PLWHA treated in facilities located outside the metropolitan region (TR = 1.09; 95% CI: 1.05; 1.12) and in facilities serving <50 PLWHA (TR = 1.15; 95% CI: 1.09; 1.21).

**Table 1. Descriptive analysis and bivariate analysis of the time from treatment initiation to VS, according to individual variables.** Qualiaids-Brazil Cohort, 2015–2018 (N = 128,360).

| Variable | Total | % | | Bivariate | | |
|---|---|---|---|---|---|---|
| | | | Suppression* | Time Ratio | LT | HT |
| **Sex** | | | | | | |
| Male | 89,586 | 69.8 | 89.5 | ref | | |
| Female | 38,774 | 30.2 | 88.9 | 0.93 | 0.93 | 0.94** |
| **Age group (in years)** | | | | | | |
| 15–19 | 5,761 | 4.5 | 87.1 | 1.04 | 1.02 | 1.07** |
| 20–29 | 44,823 | 34.9 | 89.7 | 1.01 | 0.99 | 1.02 |
| 30–39 | 37,672 | 29.4 | 88.8 | 1.03 | 1.02 | 1.05** |
| 40–49 | 23,214 | 18.1 | 89.2 | 1.03 | 1.02 | 1.05** |
| 50–59 | 12,081 | 9.4 | 90.4 | ref | | |
| ≥60 | 4,809 | 3.8 | 91.3 | 0.98 | 0.96 | 1.01 |
| **Race/skin color** | | | | | | |
| White | 57,183 | 44.5 | 91.0 | ref | | |
| Black | 14,181 | 11.1 | 87.8 | 1.04 | 1.02 | 1.05** |
| Yellow | 953 | 0.7 | 88.9 | 1.03 | 0.98 | 1.09 |
| Brown | 55,772 | 43.5 | 88.2 | 1.04 | 1.03 | 1.05** |
| Indigenous | 271 | 0.2 | 79.3 | 1.22 | 1.11 | 1.34** |
| **Education (in years of schooling)** | | | | | | |
| None | 2,278 | 1.8 | 86.7 | 1.18 | 1.14 | 1.22** |
| 1–3 | 11,132 | 8.7 | 85.6 | 1.21 | 1.19 | 1.24** |
| 4–7 | 34,298 | 26.7 | 86.9 | 1.17 | 1.16 | 1.19** |
| 8–11 | 52,042 | 40.5 | 90.1 | 1.07 | 1.06 | 1.08** |
| ≥12 | 28,610 | 22.3 | 92.5 | ref | | |
| **Initial therapeutic regimen** | | | | | | |
| Preferred regimen (2017–2018: NNRTI+1INI) | 48,410 | 37.7 | 89.6 | ref | | |
| Preferred regimen (2015–2016: 2NRTI + 1NNRTI) | 79,288 | 61.8 | 89.3 | 1.54 | 1.53 | 1.55** |
| Authorized special regimens | 118 | 0.1 | 83.1 | 1.39 | 1.21 | 1.60** |
| Unauthorized regimens | 544 | 0.4 | 85.7 | 1.48 | 1.39 | 1.58** |
| **Initial CD4 lymphocyte count (cells/mm³)** | | | | | | |
| <200 | 33,327 | 26.0 | 82.7 | 1.33 | 1.31 | 1.35** |
| 200–349 | 25,054 | 19.5 | 89.1 | 1.14 | 1.13 | 1.15** |
| 350–499 | 26,329 | 20.5 | 91.6 | 1.07 | 1.05 | 1.08** |
| ≥500 | 43,650 | 34.0 | 93.2 | ref | | |
| **Initial VL count (copies/mL)** | | | | | | |
| ≤100,000 | 117,864 | 91.8 | 91.9 | ref | | |
| >100,000 | 10,496 | 8.2 | 82.6 | 1.39 | 1.38 | 1.40** |
| **Active tuberculosis episode until suppression** | | | | | | |
| No | 122,014 | 95.1 | 90.2 | ref | | |
| Yes | 6,346 | 4.9 | 72.9 | 1.45 | 1.42 | 1.48** |
| **Therapeutic regimen change before suppression** | | | | | | |
| No | 124,305 | 96.8 | 89.0 | ref | | |
| Yes | 4,055 | 3.2 | 100.0 | 1.45 | 1.42 | 1.49** |
| **Adherence** | | | | | | |
| ≥95% | 54,618 | 42.5 | 92.6 | Ref | | |
| 80–95% | 66,334 | 51.7 | 89.4 | 1.40 | 1.39 | 1.41** |

(*Continued*)

**Table 1.** (Continued)

| Variable | Total | % | | Bivariate | | |
|---|---|---|---|---|---|---|
| | | | Suppression* | Time Ratio | LT | HT |
| <80% | 7,408 | 5.8 | 65.4 | 2.17 | 2.12 | 2.21** |

* Chi-square p<0.05 for all variables

**P≤0.001

The variables were then analyzed in the multiple models (Table 3). The time ratios of individual and facility-related variables in the multiple models followed the same direction as in the bivariate models.

Among individual variables, the following were associated with longer time to VS: male sex; age below 50 years; Black and Brown, race/skin color; education ≤11 years of schooling; therapeutic regimen 2NRTI + 1NNRTI; authorized special regimens or unauthorized regimens; initial CD4 count <500 cells/mm$^3$; initial VL >100,000 copies/mL; active tuberculosis episode; therapeutic regimen change; and treatment adherence <95%. Regarding the characteristics of treatment facilities, PLWHA who received care in facilities in the North, Northeast, and Central-West regions had a longer time to VS. The same was observed for PLWHA assisted by facilities located outside the metropolitan region (TR = 1.06; 95% CI: 1.02; 1.10) and by facilities attending <50 PLWHA (TR = 1.15; 95% CI: 1.05; 1.26).

Table 4 shows the difference in unexplained variability for each model. Model 2 showed the lowest unexplained variability, which is consistent with its higher number of explanatory variables. The model adjusted for facility-related variables (model 3) exhibited the lowest ICC (7.7%), indicating that this adjustment was able to reduce heterogeneity in time to VS. Furthermore, it was observed that 16.2% of the variability in time to VS was explained by individual sociodemographic and clinical variables (model 2); however, 13.2% of this variability can be explained solely by the facility-related variables. The log likelihood indicated that model 2 was the best fit.

The results of the sensitivity analyses were consistent with the findings, showing small variations in effect estimates (S2 and S3 Tables).

**Table 2. Bivariate analysis of the time between treatment initiation and VS, according to facility-related variables.** Qualiaids-Brazil Cohort, 2015–2018 (N = 128,360).

| Variable | TR (95% CI) |
|---|---|
| *Geographic region* | |
| Central-West | 1.10[1.03;1.18] |
| North | 1.13[1.06;1.21] |
| Northeast | 1.14[1.09;1.19] |
| South | 0.99[0.96;1.04] |
| Southeast | Ref |
| *Facility location (municipality)* | |
| Metropolitan region | Ref |
| Other | 1.09[1.05;1.12] |
| *Number of patients served* | |
| ≤50 | 1.15[1.09;1.21] |
| 51–500 | 1.03[0.98;1.07] |
| >500 | Ref |

**Table 3. Results of hierarchically adjusted multilevel models for time to VS, according to individual and facility-related characteristics, Qualiaids-Brazil Cohort, 2015–2018 (N = 128,360).**

| | | | Variable | TR (95% CI) |
|---|---|---|---|---|
| Model 1 | Model 2 | | *Sex* | |
| | | | Male | ref |
| | | | Female | 0.91[0.90;0.92] |
| | | | *Age group (in years)* | |
| | | | 15–19 | 1.14[1.11;1.16] |
| | | | 20–29 | 1.08[1.07;1.10] |
| | | | 30–39 | 1.06[1.04;1.07] |
| | | | 40–49 | 1.03[1.02;1.05] |
| | | | 50–59 | Ref |
| | | | ≥60 | 0.97[0.95;1.00] |
| | | | *Race/skin color* | |
| | | | White | Ref |
| | | | Black | 1.04[1.02;1.05] |
| | | | Yellow | 1.01[0.97;1.06] |
| | | | Brown | 1.03[1.02;1.04] |
| | | | Indigenous | 1.17[1.07;1.27] |
| | | | *Education (in years of schooling)* | |
| | | | None | 1.18[1.14;1.21] |
| | | | 1–3 | 1.20[1.18;1.22] |
| | | | 4–7 | 1.15[1.14;1.16] |
| | | | 8–11 | 1.06[1.05;1.07] |
| | | | ≥12 | Ref |
| | | | *Initial therapeutic regimen* | |
| | | | Preferred regimen (2017–2018: NNRTI+1INI) | Ref |
| | | | Preferred regimen (2015–2016: 2NRTI + 1NNRTI) | 1.33[1.32;1.34] |
| | | | Authorized special regimens | 1.07[0.94;1.21] |
| | | | Unauthorized regimens | 1.23[1.16;1.31] |
| | | | *Initial CD4 lymphocyte count (cells/mm$^3$)* | |
| | | | ≤200 | 1.09[1.08;1.10] |
| | | | 200–349 | 1.09[1.08;1.10] |
| | | | 350–499 | 1.04[1.03;1.05] |
| | | | ≥500 | Ref |
| | | | *Initial VL count (copies/mL)* | |
| | | | ≤100,000 | Ref |
| | | | >100,000 | 1.19[1.17;1.20] |
| | | | *Active tuberculosis episode until suppression* | |
| | | | No | Ref |
| | | | Yes | 1.10[1.08;1.12] |
| | | | *Therapeutic regimen change* | |
| | | | No | Ref |
| | | | Yes | 1.15[1.13;1.18] |
| | | | *Adherence* | |
| | | | ≥95% | Ref |
| | | | 80–95% | 1.19[1.18;1.21] |
| | | | <80% | 1.83[1.79;1.86] |

*(Continued)*

**Table 3.** (Continued)

| | | | Variable | TR (95% CI) |
|---|---|---|---|---|
| | | | *Geographic region* | |
| | | | Central-West | 1.18[1.08;1.28] |
| | | | North | 1.20[1.11;1.30] |
| | | | Northeast | 1.20[1.14;1.16] |
| | | | South | 1.01[0.97;1.06] |
| | | | Southeast | Ref |
| | | | *Facility location (municipality)* | |
| | | | Metropolitan region | Ref |
| | | | Other | 1.06[1.02;1.10] |
| | | | *Number of patients served* | |
| | | Model 3 | ≤50 | 1.15[1.05;1.26] |
| | | | 51–500 | 0.96[0.89;1.04] |
| | | | >500 | Ref |

Model 1 – sociodemographic characteristics; Model 2 – sociodemographic + clinical characteristics; Model 3 – facility-related characteristics.

All models were adjusted for the number of VL tests performed until VS.

## Discussion

Among PLWHA participating in the Qualiaids-Brazil Cohort who initiated ART in SUS healthcare facilities between 2015 and 2018, nearly 90% achieved VS, half of whom did so within five months of treatment.

Individuals in the following categories had a higher proportion of VS during the cohort's time period: male sex, aged 50–59 years, white race/skin color, with ≥12 years of education, initial treatment regimen NNRTI+1INI, initial CD4 lymphocyte count ≥500 cells/mm$^3$, initial VL count ≤100,000 copies/mL, without an active tuberculosis episode, without therapeutic regimen change, with ≥95% adherence to treatment, and receiving clinical follow-up in facilities located in the Southeast or South regions, in metropolitan areas, and in facilities serving >500 PLWHA.

Individual variables accounted for most of the variability in time to VS; however, facility-related variables also played a noteworthy role.

Regarding individual characteristics, there was an association with longer time to VS of individuals of brown and black race/skin color, as well as those with lower education levels. In Brazil, these characteristics are proxies for poorer social conditions [52]. This finding is consistent with several international studies on ethnic and social disparities [53–56].

**Table 4. Results of the random-effects analysis of models null through 3, Qualiaids-Brazil Cohort, 2015–2018 (N = 128,360).**

| | Null Model | Model 1 | Model 2 | Model 3 |
|---|---|---|---|---|
| Random Variance (SE) | 0.068 (0.004) | 0.066 (0.004) | 0.057 (0.003) | 0.059 (0.003) |
| ICC | 0.093 | 0.091 | 0.083 | 0.077 |
| PCV | ref | 0.029 | 0.162 | 0.132 |
| Log Likelihood | -712903.7 | -712188.3 | -706630.4 | -712830.4 |

SE: standard error; ICC: intraclass correlation coefficient; PCV: proportional change in variance

Social issues are intertwined with clinical issues. The study demonstrated that late diagnosis, with a low CD4 level (<200) and very high VL (>100,000), delays time to VS, echoing numerous studies that associate these factors with negative outcomes. Since 2015, Brazil has maintained a late diagnosis proportion of around 25% [8]. The trends observed in countries with similar epidemic profiles also confirm the maintenance of late diagnosis proportions. Globally, it is still observed that the proportion of people diagnosed late is changing little as the effectiveness of ART continues to improve [10, 57–59].

In Brazil, especially, the second most important clinical condition associated with negative outcomes is tuberculosis (TB). This situation is similar to that of low-income countries, where TB is the leading cause of death among PLWHA, notably in the first year after diagnosis [60, 61]. Brazil is among the 12 countries with the highest global TB burden. Regarding coinfection HIV and tuberculosis, it has the highest burden of those countries [62]. Coinfection with TB is associated with lower CD4 levels and higher VL levels, underscoring the importance of HIV treatment facilities in diagnosing and treating both latent and active tuberculosis [63, 64].

The negative impact on time to VS resulting from poor adherence to HIV treatment reaffirms numerous studies that, since the HIV treatment became available, have emphasized the crucial role of promotion, monitoring, and support by treatment facilities for those with difficulty in adhering to the prescribed regimen [65, 66].

The longer time to achieving VS among non-white individuals with lower education levels follows the general trend highlighted in studies of mortality, late diagnosis, loss to follow-up, and tuberculosis, reflecting social and ethnic inequality in the country. Despite significant progress in addressing the HIV epidemic through free and universal access to high-effectiveness care and medication across the country, the healthcare system has not successfully mitigated inequalities to achieved desired health outcomes, reflecting disparities in the allocation and organization of treatment facilities. The higher concentration of facilities in the Southeast and South regions and in metropolitan areas corresponds to the greater concentration of PLWHA in these locations. Certainly, there is also a need for smaller facilities, particularly in smaller cities within the regions. On the other hand, the allocation of facilities in both large and small municipalities for a small number of patients (<50) appears to be problematic, suggesting significant diversity of HIV care policies among municipal administrations [28, 67]. While certain geographical conditions may warrant the establishment of small-scale facilities, evidence repeatedly shows a higher likelihood of negative outcomes in facilities with a reduced number of patients [21, 68]. Balancing easy access to treatment with professional training and organization therefore is an important priority to ensure appropriate quality of care. Disparities in the organization of HIV care at regional, district, and even municipal levels have been highlighted in national studies [31–34, 69, 70].

The study revealed a significant flaw in the quality of the care provided by treatment facilities: among PLWHA who achieved VS during the study period, 67.2% did not receive VL testing within the timeframe specified by the protocol (66 days) [38]. The deficiency in monitoring has been pointed out in studies of the records of two populous Brazilian states, Minas Gerais [24] and Rio de Janeiro [47]. The issue appears to be predominantly organizational in nature rather than caused by supply of tests since data indicate high and uniform availability of VL tests across the country's public service network [33, 34]. On the other hand, the request for VL tests and scheduling of follow-up appointments, lacks managerial mechanisms, such as electronic control of appointment scheduling and tests rather than relying on individual physician's request tests. We suggested that this flaw in monitoring, translated into longer-than-recommended intervals between tests, stems from the high demand for medical appointments—a common issue in larger facilities—thus influencing physicians' decisions regarding test and appointment scheduling. Another possibility is that the longer intervals

may be a recommendation from facility managers to alleviate the pressure of demand for appointments. Detailed studies focusing on the organizational processes in treatment facilities are needed to support interventions capable of improving this situation as quickly as possible.

Promptness in having the first VL test after treatment initiation is fully justified by the clinical need for timely detection of viral failures and adherence issues [1, 71]. It may also operate as an incentive to maintain adherence for those who achieve rapid VS [72, 73]. Proper VL monitoring and rapid VS rate are indicators of quality of care [12].

The ability to quickly achieve VS is feasible given the effectiveness of available ART regimen, especially for those who initiate treatment with CD4>200 and low VL and maintain good adherence. ART effectiveness has increased with widespread use of regimens containing integrase inhibitors, especially dolutegravir [74], which is part of the recommended regimen for treatment initiation in Brazil. There is also repeated evidence that rapid treatment initiation accelerates time to VS [75, 76]. However, the time to initiation could not be addressed in this study.

The median time to VS was 55 days for those who had the first VL test after starting treatment within the established timeframe (66 days). The median is similar to that found in observational studies conducted in locations with a comparable epidemic profile to Brazil's (concentrated epidemic) and with free access to treatment, such as Atlanta, 2016 (57 days) [77], and London, 2018 (56 days) [78].

Considering the importance of rapid suppression, Xia [54] and Dombrowski [12] suggested that the treatment time parameter for estimating the "suppressed" bar of the cascade—usually set at 12 months or, as in Brazil, at six months—be reduced to three months. They argue that rapid initiation and the high efficacy of current ART shorten this time, as exemplified by studies that, using a VL threshold of <200 copies/mL in three months, found VS proportions of 37% [54] and 45% [17]. With a lower threshold of <50 copies/mL, the proportion of those who achieved VS, in this study, within the shortest evaluated interval (66 days—27.5%) would likely be higher if the monitoring protocol had been adhered to for most PLWHA. Among all those who achieved VS and were tested within the first three months, VS rate was 36.1%, indicating that it would be possible in Brazil to shorten the threshold to three months, even as an incentive for improving VL monitoring.

This study was the first nationwide cohort, based on a longitudinal database, that analyzed time to VS in PLWHA receiving treatment in the SUS. It was also one of the largest operational cohorts in the international literature. The inclusion of the characteristics of healthcare facilities in the analyses reinforces the study's originality.

As time to VS may be influenced by the number of VL tests conducted during the period, the inclusion of this variable in the analyses is also a strength to be noted.

Nonetheless, there are were limitations to our analysis. First, in the studied database there was no explicit definition of the healthcare system in which PLWHA received care. Thus, a previously published standardized classification strategy, as defined in the methods section, was employed to minimize that limitation. It should also be noted that the use of secondary sources such as developed for operational purposes, involves inconsistencies and missing data, which may lead to misclassification errors and underestimation of the measures of interest. However, the definition of inclusion criteria sensitive to potential inconsistencies and the imputation of data for relevant variables reduces the possibility of interference in the results [29].

This study of a large cohort of PLWA in Brazil was the first to analyze time to viral suppression and the first to include characteristics of healthcare facilities. The results strongly suggest the need for more comprehensive longitudinal studies, including more individual and health facilities characteristics, that can further contribute to healthcare policies for PLWHA in Brazil.

While acknowledging the limitations, the findings seem sufficient to indicate the urgency of improving VL monitoring in the treatment facilities network across the country. This must be carried out as part of a coordinated effort to provide more equitable access and capacity for HIV facilities to optimize quality care and reduce time to viral suppression.

## Supporting information

**S1 Table. Time between ART initiation and VL testing in the first six months of treatment.** Qualiaids-Brazil Cohort, 2015–2018 (N = 101,822).
(DOCX)

**S2 Table. Results of hierarchically adjusted multilevel models for time to VS, as well as individual and facility-related characteristics, restricted to those with information on exposure category.** Qualiaids-Brazil Cohort, 2015–2018 (N = 84,747).
(DOCX)

**S3 Table. Results of hierarchically adjusted multilevel models for time to VS, according to individual and facility-related characteristics WITHOUT adjustment for number of VL tests.** Qualiaids-Brazil Cohort, 2015–2018 (N = 84,747).
(DOCX)

## Author Contributions

**Conceptualization:** Maria Ines Battistella Nemes, Ana Paula Sayuri Sato, Ana Maroso Alves.

**Data curation:** Maria Ines Battistella Nemes.

**Formal analysis:** Ana Paula Sayuri Sato, Barbara Reis-Santos, Ana Maroso Alves, Felipe Parra do Nascimento.

**Investigation:** Maria Ines Battistella Nemes.

**Methodology:** Maria Ines Battistella Nemes, Ana Paula Sayuri Sato.

**Supervision:** Maria Ines Battistella Nemes.

**Validation:** Ana Paula Sayuri Sato, Ana Maroso Alves, Felipe Parra do Nascimento.

**Visualization:** Ana Paula Sayuri Sato, Barbara Reis-Santos, Ana Maroso Alves, Felipe Parra do Nascimento.

**Writing – original draft:** Maria Ines Battistella Nemes, Ana Paula Sayuri Sato, Barbara Reis-Santos, Ana Maroso Alves, Felipe Parra do Nascimento, Bruce Agins.

**Writing – review & editing:** Maria Ines Battistella Nemes, Ana Paula Sayuri Sato, Barbara Reis-Santos, Ana Maroso Alves, Felipe Parra do Nascimento, Bruce Agins.

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
