## [Decision Letter · Decision Letter 0]

11 Jul 2024

Time from treatment initiation to HIV viral suppression in public care facilities  in Brazil: a nationwide linked databases cohort

PONE-D-24-20580

Dear Dr. Nemes,

We’re pleased to inform you that your manuscript has been judged scientifically suitable for publication and will be formally accepted for publication once it meets all outstanding technical requirements.

Kind regards,

Angelica Espinosa Miranda, M.D., Ph.D.

Academic Editor

PLOS ONE

 [CNPq (Conselho Nacional de Desenvolvimento Científico e Tecnológico | 

National Council for Scientific and Technological Development) - U$D 9,600.00 (MIBN)

https://www.gov.br/cnpq/pt-br

OPAS (Organização Pan-Americana da Saúde - Organização Mundial da Saúde - Região das Américas| Pan American Health Organization - World Health Organization - Americas Region)- U$D 31,670.00 (MIBN)

https://www.paho.org/en

USP (Universidade de São Paulo | University of São Paulo) – U$D 6,360.00 (APSS)

https://www5.usp.br/].  

Please respond by return e-mail so that we can amend your financial disclosure and competing interests on your behalf.

4. Please remove your figures from within your manuscript file, leaving only the individual TIFF/EPS image files, uploaded separately. These will be automatically included in the reviewers’ PDF.

5. Please upload a copy of Supplementary table 1, Supplementary table 2 and Supplementary table 3 to which you refer in your text on page 11 and 17. Please amend the file type to 'Supporting Information'. If the Supplementary file is no longer to be included as part of the submission please remove all reference to it within the text.

Additional Editor Comments:

The article is a well-designed study with adequate methods, presenting results in a clear and concise manner. The discussion  addresses both the broader implications and the limitations of the study. The data offer valuable insights for public health policies in Brazil and can be instrumental in planning and implementing these policies.

Reviewers' comments:

Reviewer's Responses to Questions

**Comments to the Author**

1. Is the manuscript technically sound, and do the data support the conclusions?

Reviewer #1: Yes

Reviewer #2: Yes

2. Has the statistical analysis been performed appropriately and rigorously? 

Reviewer #1: Yes

Reviewer #2: Yes

3. Have the authors made all data underlying the findings in their manuscript fully available?

Reviewer #1: Yes

Reviewer #2: Yes

4. Is the manuscript presented in an intelligible fashion and written in standard English?

Reviewer #1: Yes

Reviewer #2: Yes

5. Review Comments to the Author

Reviewer #1: The quality of the manuscript is very high and the methodology is appropriate, considering an observational study and a national database. The results are consistent with other published articles. However, there are limitations to measuring treatment adherence because it is based on the date of dispensation. Some information such as tuberculosis, race/color, sexual orientation and education level depend of filling out a specific antiretroviral form in general by a health worker with only a little time to complete it and can determine a bias of this information. Presence of tuberculosis and race/color was associated with outcomes. A longer time until viral load supression was associated with antiretroviral regimen with a low genetic barrier, however there is no report as to whether a genotyping test was performed before the start of treatment and transmited resistance could also determine bias.

Reviewer #2: The presented work is scientifically sound and addresses an important issue, the influence of characteristics of HIV health care clinics to time from ART initiation to VS. It is a nationwide study, with strong potential to positively influence stakeholders and decision makers to foster strategies to improve HIV continuum of care in Brazil.

6. PLOS authors have the option to publish the peer review history of their article (what does this mean?). If published, this will include your full peer review and any attached files.

Reviewer #1: **Yes: **Ronaldo Campos Hallal

Reviewer #2: No

---

## [Editor Report · Acceptance letter]

25 Jul 2024

PONE-D-24-20580 

PLOS ONE

Dear Dr. Nemes, 

I'm pleased to inform you that your manuscript has been deemed suitable for publication in PLOS ONE. Congratulations! Your manuscript is now being handed over to our production team.

Kind regards, 

on behalf of

Dr. Angelica Espinosa Miranda 

Academic Editor

PLOS ONE